# Indels allow antiviral proteins to evolve functional novelty inaccessible by missense mutations

## Graphical abstract

## Authors

Jeannette L. Tenthorey,
Serena del Banco, Ishrak Ramzan,
Hayley Klingenberg, Chang Liu,
Michael Emerman, Harmit S. Malik

## Correspondence

jeannette.tenthorey@ucsf.edu

## In brief

Tenthorey et al. compare the effects of missense and indel mutations on the acquisition of functional novelty by the rapidly evolving antiviral protein TRIM5α. They find that single indel mutations allow human TRIM5α to restrict a virus that otherwise requires more than five substitutions, revealing the evolutionary potential of indels.

## Highlights

- The antiviral protein TRIM5α evolves rapidly, sampling missense and indel mutations

- Naturally occurring indels allow TRIM5α to defend against new retroviruses

- We compare the evolutionary potential of all possible substitutions and indels

- Indels allow TRIM5α to restrict a virus that otherwise requires >5 substitutions

 Tenthorey et al., 2025, Cell Genomics 5, 100818
June 11, 2025 © 2025 The Authors. Published by Elsevier Inc.

## Article

# Indels allow antiviral proteins to evolve functional novelty inaccessible by missense mutations

Jeannette L. Tenthorey,[1,6,*] Serena del Banco,[2,5] Ishrak Ramzan,[1] Hayley Klingenberg,[1] Chang Liu,[1] Michael Emerman,[2,3] and Harmit S. Malik[2,4]

[1]Cellular and Molecular Pharmacology Department, University of California, San Francisco, San Francisco, CA 94158, USA
[2]Division of Basic Sciences, Fred Hutchinson Cancer Center, Seattle, WA, USA
[3]Division of Human Biology, Fred Hutchinson Cancer Center, Seattle, WA, USA
[4]Howard Hughes Medical Institute, Fred Hutchinson Cancer Center, Seattle, WA, USA
[5]Present address: Molecular and Cellular Biology Graduate Program, University of Washington, Seattle, WA, USA
[6]Lead contact
*Correspondence: jeannette.tenthorey@ucsf.edu

## SUMMARY

Antiviral proteins often evolve rapidly at virus-binding interfaces to defend against new viruses. We investigated whether antiviral adaptation via missense mutations might face limits, which insertion or deletion mutations (indels) could overcome. Using high-throughput saturation missense mutagenesis, we identify one such case of a nearly insurmountable evolutionary challenge: the human anti-retroviral protein TRIM5α requires more than five missense mutations in its specificity-determining v1 loop to restrict a divergent simian immunodeficiency virus (SIV). However, through a novel saturating indel scanning methodology, we find that duplicating just one amino acid in v1 enables human TRIM5α to potently restrict SIV in a single evolutionary step. Moreover, natural primate TRIM5α v1 loops have evolved indels that confer novel antiviral specificities. Thus, indels enable antiviral proteins to overcome viral challenges otherwise inaccessible by missense mutations. Our findings reveal the potential of often-overlooked indel mutations in driving protein innovation.

## INTRODUCTION

Host genomes deploy restriction factors—innate immune proteins with direct antiviral activity—to defend against viral infections.[1] Mutations in restriction factors' virus-binding interfaces can dramatically alter their viral specificities. As viruses escape restriction factor defenses or new viruses enter the population, they drive selection for novel host antiviral variants in a process likened to an evolutionary arms race.[2] As a result, restriction factors evolve rapidly either to confer recognition of new viruses[3–5] or to escape binding by virus-encoded antagonists.[6–8] Because of their potentially profound effects on protein-protein interactions,[9] considerable effort has been devoted to the functional consequences of missense mutations in host-virus evolutionary arms races and in protein evolution more broadly.[10,11] Such approaches have included targeted missense mutations at sites that recurrently undergo amino acid substitutions[4,12,13] and random mutagenesis via deep mutational scanning (DMS).[10] Both approaches have been successful at eliciting missense mutations in restriction factors that gain or enhance restriction against specific viruses.

After single-nucleotide substitutions, insertion or deletion mutations (hereafter referred to as indels) are the second most common mutational process, accounting for 10%–20% of human genetic variation.[14] There is strong purifying selection to remove both frame-shifting and in-frame (multiples of 3 nucleotides) indels from protein-coding sequences.[15,16] Frameshift indels dramatically alter downstream protein sequences and are likely to be instantaneously deleterious, but even in-frame indels often introduce register shifts or bulges in secondary structural elements that destabilize protein folds.[17,18] In head-to-head comparisons, indels exert much more detrimental effects on protein function than missense mutations.[19–21] The propensity of indels to break protein-protein interactions has been exploited by host proteins, which have evolved indels to escape viral entry[22] or viral antagonism.[6,13] Yet, how indel variants in restriction factors might gain *de novo* functions, including enhanced antiviral specificity via increased protein-protein interaction, has not been systematically tested or compared with missense mutations.

Here, we directly compare the propensity of missense vs. indel mutations to confer functional novelty upon human TRIM5α, a restriction factor that inhibits retroviral infection.[23] TRIM5α uses its disordered v1 loop to bind the capsid core of incoming lentiviruses, HIV-related retroviruses circulating in hominoid and Old World monkey species, although the molecular details of this interface remain unclear.[24–26] The v1 loop is a small segment (typically 22 amino acids) of the virus-binding B30.2 domain of TRIM5α, which also contains self-oligomerization

domains (coiled coil and B-box) and a ubiquitin ligase (RING) domain within the ~500 amino acid protein.[27] The name v1 (for "variable region 1") reflects its dramatic divergence across primate TRIM5α proteins, both in sequence (0%–95% identity among primate pairs compared to ≥70% identity across the entire protein) and length (13–42 amino acids)[3,28] (see Data S5). Previous experiments revealed that many single missense mutations in its rapidly evolving v1 loop, including an R332P substitution, enable human TRIM5α to gain restriction of HIV-1 and other lentiviruses.[12,28] In contrast, here we explore a case where single point mutations in human TRIM5α cannot confer protection against a simian lentivirus (simian immunodeficiency virus [SIV]sab) endemic to sabaeus monkeys (*Chlorocebus sabaeus*).[26] Using two high-throughput methods to evaluate the antiviral function of missense variant libraries of TRIM5α, we find that SIVsab represents a nearly insurmountable challenge for TRIM5α to overcome using missense mutations; human TRIM5α must acquire specific missense mutations at multiple positions to gain SIVsab restriction. In contrast, we find that a simple indel mutation is sufficient for human TRIM5α to acquire SIVsab inhibition in a single evolutionary step. Thus, indel mutations allow host restriction factors to traverse otherwise impassable fitness landscapes in evolutionary arms races with viruses.

## RESULTS

### Human TRIM5α cannot acquire SIVsab restriction through missense mutations

To explore how TRIM5α evolves new antiviral functions, we focused our analysis on its rapidly evolving v1 loop, which previous studies identified as the critical determinant of specificity for lentiviruses.[3] For example, while human TRIM5α has little anti-lentiviral activity,[29] swapping its entire v1 loop for that of the rhesus macaque homolog, which potently restricts several lentiviruses,[29] is sufficient to confer these antiviral functions upon human TRIM5α.[3] Even a single rhesus-mimicking missense mutation from this v1 loop (R332P) confers human TRIM5α with enhanced protection against HIV-1 and many other lentiviruses.[12,28] However, we found that this mutation does not alter human TRIM5α restriction of SIVsab (Figure 1A). We hypothesized that distinct missense mutations might confer human TRIM5α with anti-SIVsab function. To identify such mutations, we generated a DMS library of all possible single missense variants in the v1 loop of full-length human TRIM5α (Figure 1B). We expressed this TRIM5α v1 DMS library in cells that are naturally deficient for TRIM5α[30] by low-dose viral transduction, such that each cell expresses a single TRIM5α variant. We then infected the cellular library with single-cycle (self-inactivating after one infection cycle) SIVsab virions encoding GFP, sorted cells that remained GFP[neg] even after two consecutive challenges with SIVsab-GFP, and used deep sequencing to identify their TRIM5α variants. We used the enrichment of TRIM5α variants in the final sorted pool as a quantitative proxy for enhanced antiviral function against SIVsab, as previously validated.[28]

In a previous analysis of human TRIM5α DMS variants, we found that at least half of all possible single missense mutations in the v1 loop enhance HIV-1 restriction.[28] In stark contrast, the same library and screening workflow revealed that no missense

variants enhanced SIVsab restriction (Figure 1C). Indeed, multiple TRIM5α single missense variants that gain restriction of HIV-1[28] cannot restrict SIVsab (Figure S1). Although we initially identified 3 slightly enriched variants (relative to nonfunctional nonsense mutants), subsequent experiments found that these missense mutations did not allow human TRIM5α to inhibit SIVsab, either alone or in combination (Figure 1D). Thus, their modest enrichment scores represent noise rather than true enhancement of SIVsab restriction. We conclude that single missense mutations in the lentiviral-specificity-determining v1 loop of human TRIM5α cannot confer antiviral function against SIVsab, which poses a more formidable evolutionary challenge than several other lentiviruses.

Our findings raised the possibility that, unlike for HIV-1, the TRIM5α v1 loop may not be a critical determinant of SIVsab restriction. We tested this possibility by swapping the v1 loop between human and rhesus TRIM5α, which can potently inhibit SIVsab (Figure 2A). Swapping the v1 loop from rhesus TRIM5α into the human protein greatly enhances SIVsab restriction, although not to the same level as full-length rhesus TRIM5α (Figure 2A). The partial functionality of this chimera is consistent with its modest activity (relative to rhesus TRIM5α) against HIV-1.[3] Conversely, a reciprocal swap of the human v1 into rhesus TRIM5α leads to a complete loss of anti-SIVsab function. These results indicate that the v1 loop indeed dictates recognition of SIVsab, as it does for HIV-1.

These chimeras also indicated that human TRIM5α is able to acquire SIVsab restriction via the 9 differences (8 missense mutations and one 2-amino-acid insertion) between the human and rhesus v1 loop. To determine the minimal number of these mutations required to achieve SIVsab restriction, we screened a library of human TRIM5α variants bearing either the human or rhesus residue at each of the 9 variable v1 positions; in so doing, we sampled all possible evolutionary intermediates between the human and rhesus v1 loop (Figure 2B). Of these $2^9$ combinatorial variants, only 13 (4% of 349 analyzed variants; see STAR Methods: TRIM5α variant library construction for a DNA synthesis error resulting in excluded variants) were strongly enriched in the SIVsab-restrictor pool (Figure 2C). Each of these SIVsab restrictors encoded the rhesus TRIM5α residue at 5 central positions in the v1 loop, corresponding to residues 332–337 in human TRIM5α (Figures 2C, 2D, and S2A). These 5 rhesus-mimicking mutations, which include 4 substitutions and 1 insertion, are all necessary for SIVsab restriction; mutating any of these residues led to a loss of SIVsab restriction by a human TRIM5α chimera bearing the rhesus v1 loop (Figure 2E, "RhV1"). However, even these 5 mutations (shorthanded as the "5xRh" variant) are not sufficient to confer human TRIM5α with SIVsab restriction, as indicated by enrichment (Figure 2D) and validated by retesting an individually expressed 5xRh variant (Figure 2E). Only the addition of a sixth rhesus-mimicking mutation at position 330, 338, or 340 can confer antiviral function against SIVsab (Figures 2D and 2E). Each of these mutations had no effect on SIVsab restriction until at least 5 other mutations had been acquired (Figure S2B), indicating that no evolutionary intermediates conferred a selective advantage. The simultaneous requirement for 6 independent mutations suggests that human TRIM5α is extremely unlikely to evolve *de novo* SIVsab restriction via this evolutionary path.

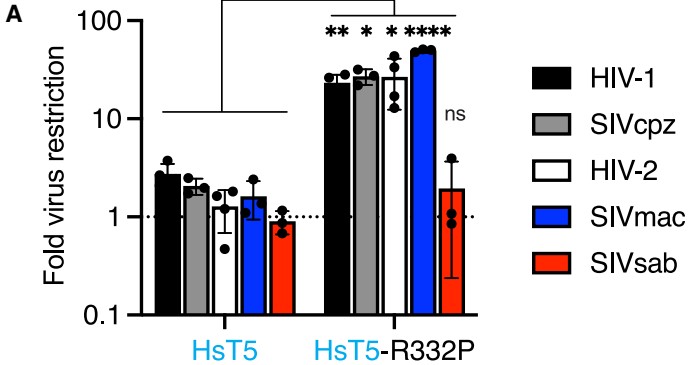

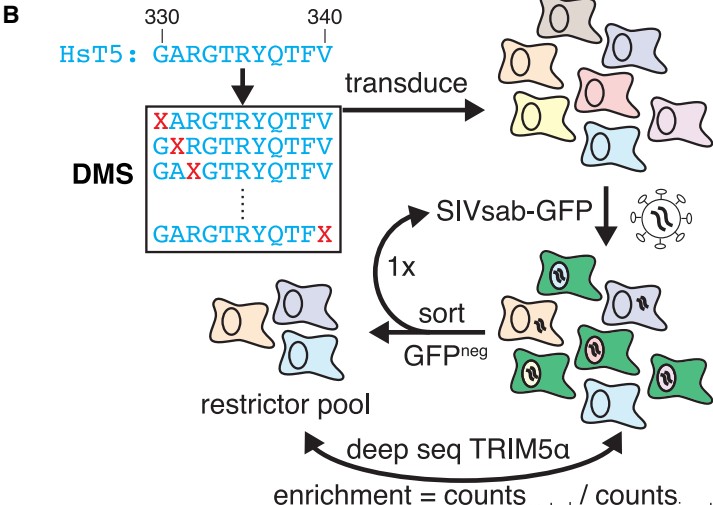

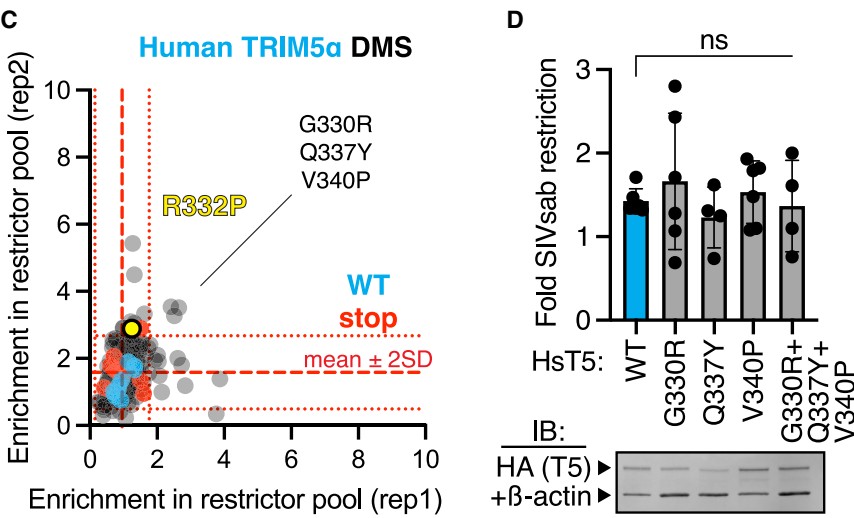

**Figure 1. Single missense mutations allow human TRIM5α to acquire function against many lentiviruses but not SIVsab**

(A) CRFK (Crandell-Rees feline kidney) cells were transduced with hemagglutinin (HA)-tagged human TRIM5α (HsT5), selected for stable expression, and then infected with GFP-expressing single-cycle viruses. Fold restriction was determined by the increase in infectious dose ($ID_5$) relative to cells expressing empty vector ($n = 3$ [SIVcpz, SIVmac, and SIVsab] or 4 [HIV-1 and HIV-2] independent experiments).

(B) A deep mutational scanning (DMS) library of HsT5, individually randomizing each v1 loop position to any amino acid (X), was stably expressed in CRFK cells, challenged with SIVsab-GFP, and fluorescence-activated cell-sorted (FACS) for uninfected $GFP^{neg}$ cells through 2 rounds of infection ($n = 2$ independent experiments). Putative gain-of-function variants were identified by deep sequencing.

(C) Missense variants (gray), including R332P (yellow), behave indistinguishably from wild-type (WT; cyan) or premature stop codon (red) variants. Only three variants were modestly enriched (>[stop mean + 2 SD] in both biological replicates).

(D) Enriched variants were individually expressed in CRFK cells and challenged with SIVsab ($n = 6$ [WT, G33R, and V340P] or 4 [Q337Y and G330R/Q337Y/V340P] independent experiments). TRIM5α protein expression was confirmed by immunoblot (IB) against the HA tag, relative to a loading control (anti-β-actin).

ns, not significant; $*p < 0.05$, $**p < 0.01$, $***p < 0.001$, and $****p < 0.0001$; (A) Welch's (HIV-1, SIVcpz, and HIV-2) or unpaired (SIVmac and SIVsab) two-tailed t test vs. WT; (D) Kruskal-Wallis one-way ANOVA with Dunn's multiple comparisons correction vs. WT. All bars, mean; error bars, SD. See also Figure S1 and Data S1.

loss-of-function rhesus TRIM5α variants (Figure S3A). We confirmed that 4 of the 5 required mutations in the human backbone (3 substitutions and 1 insertion, corresponding to human TRIM5α residues 335–337) were also strictly required in the context of rhesus TRIM5α (Figures S3B–S3D). These data indicate that the molecular requirements for SIVsab restriction are largely shared across divergent TRIM5α homologs.

Finally, we asked whether human TRIM5α might acquire SIVsab restriction via a combination of other, non-rhesus-mimicking missense mutations. Since we cannot comprehensively assay all such possible mutations in a combinatorial manner, we instead investigated how SIVsab restriction by rhesus TRIM5α, which largely shares human TRIM5α's molecular requirements (Figure S3), is affected by other, non-human-like mutations at these positions. To do so, we analyzed a DMS library of single

To determine whether the requirement for these mutations was specific to human TRIM5α, we also analyzed the human/rhesus v1 combinatorial variants in the rhesus TRIM5α backbone. Because rhesus TRIM5α potently inhibits SIVsab (Figure 2A), we sorted SIVsab-infected GFP^pos cells to identify

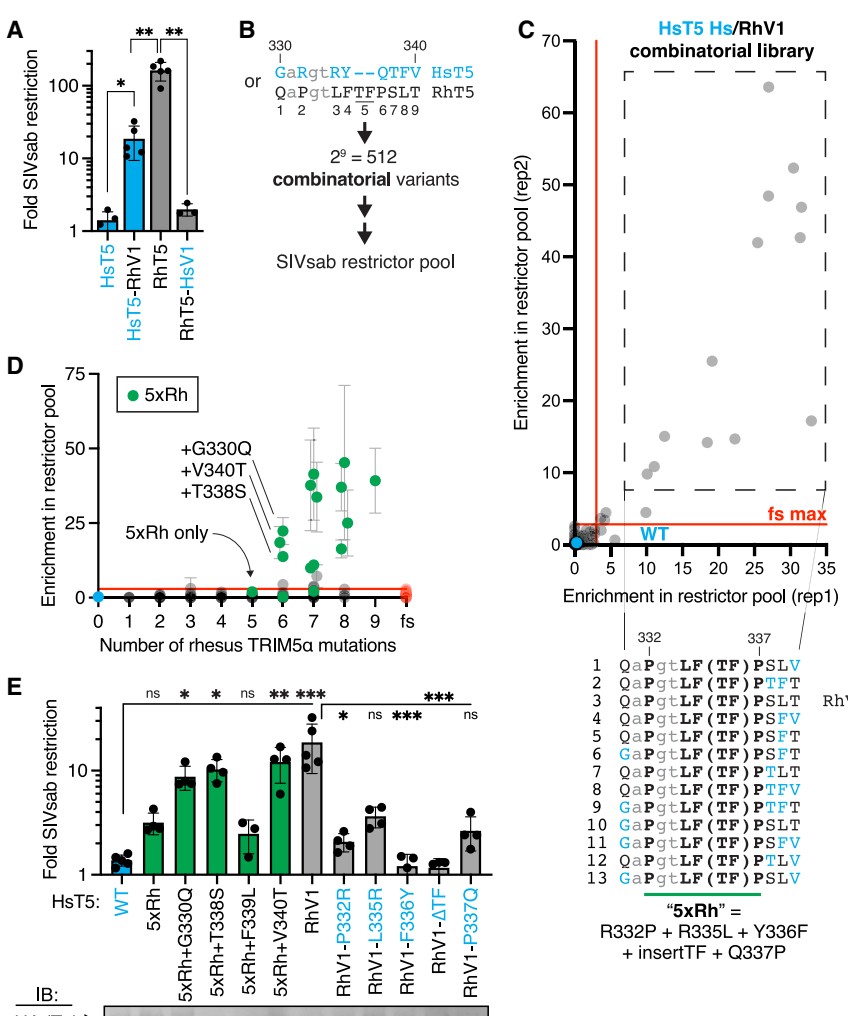

**Figure 2. Human TRIM5α requires 6 rhesus mutations to achieve SIVsab restriction**

(A) Chimeric TRIM5α constructs swapping the v1 loop (sequences in B) between human and rhesus TRIM5α (RhT5) were stably expressed in CRFK cells and challenged with SIVsab (n = 3 [HsT5 and RhT5-HsV1] or 5 [HsT5-RhV1 and RhT5] independent experiments).

(B) A combinatorial variant library, sampling either the human (cyan text) or rhesus (black text) TRIM5α residue at each v1 position, was generated in human TRIM5α to identify gain-of-function variants against SIVsab, as in Figure 1B.

(C) Ranked top hits (from 2 biological replicates) all contain at least 5 central rhesus TRIM5α mutations ("5xRh"). fs, frameshift variants (see STAR Methods).

(D) Average enrichment of all variants plotted by number of rhesus v1 mutations. Variants containing the 5xRh required mutations are highlighted (green).

(E) Human TRIM5α variants were individually expressed in CRFK cells, as confirmed by IB, and challenged with SIVsab (n = 5 [WT and RhV1], 3 [5xRh+F339L], or 4 [rest] independent experiments).

(A and E) ns, not significant; *p < 0.05, **p < 0.01, and ***p < 0.001; Welch one-way ANOVA with Dunnett's T3 multiple comparisons correction (vs. HsT5-WT, HsT5-RhV1, or RhT5, as indicated). All bars, mean; error bars, SD. See also Figures S2, S3, and Data S2.

## Human TRIM5α can acquire anti-SIVsab function in a single evolutionary step via indels

Our combinatorial mutational analysis revealed that, in addition to 4 missense changes, the 2-amino-acid insertion in the rhesus TRIM5α v1 loop was strictly

missense mutations in the rhesus TRIM5α v1 loop (Figure 3A). In previous studies, we found that most v1 loop substitutions in rhesus TRIM5α do not substantially impair HIV-1 restriction.[28] In contrast, we observed that many single missense variants of rhesus TRIM5α (54%) lost SIVsab restriction (Figures 3B and 3C). Thus, despite its antiviral potency, rhesus TRIM5α's antiviral function against SIVsab is mutationally fragile. Critically, nearly all substitutions at residues 337–342 entirely abolished SIVsab restriction (Figure 3D, compared to premature stop codon variants). These positions correspond to human TRIM5α positions 335–338, including 3 of the residues that strictly required rhesus-like substitutions to acquire SIVsab restriction. These results illustrate that only a very limited set of amino acids at each of the central v1 loop positions is compatible with anti-SIVsab function, further constraining the likelihood of human TRIM5α successfully adapting to restrict SIVsab via missense mutations. Collectively, these results indicate that SIVsab restriction poses an extraordinarily difficult challenge, which human TRIM5α cannot overcome via missense mutations alone.

necessary (Figure 2C, all restrictive variants contain +TF [T, threonine; F, phenylalanine]; Figure 2E, "RhV1ΔTF") but not sufficient (Figure S4A) for human TRIM5α to acquire rhesus-like activity against SIVsab. Restriction factors like TRIM5α frequently evolve in-frame indel mutations at or adjacent to sites undergoing rapid missense mutational evolution.[3,4,6,8,13] Therefore, we considered the possibility that indel mutations might be necessary or even sufficient for human TRIM5α to evolve antiviral activity against SIVsab. To search indel space more comprehensively, we generated a deep indel scanning (DIS) library sampling in-frame (i.e., non-frameshifting) deletions or sequence duplications from 1 to 9 amino acids long at each codon in the v1 loop (Figure 4A). Our unbiased forward evolutionary approach is designed to simulate replication slippage errors, in which DNA polymerase slips forward or backward to delete or duplicate a sequence, which accounts for ~75% of human indel mutations.[15] We limited indels to insert in phase (between codons), thereby avoiding indels that also give rise to adjacent missense mutations.[16] Although we allowed indels to extend to 9 amino acids, the vast majority of

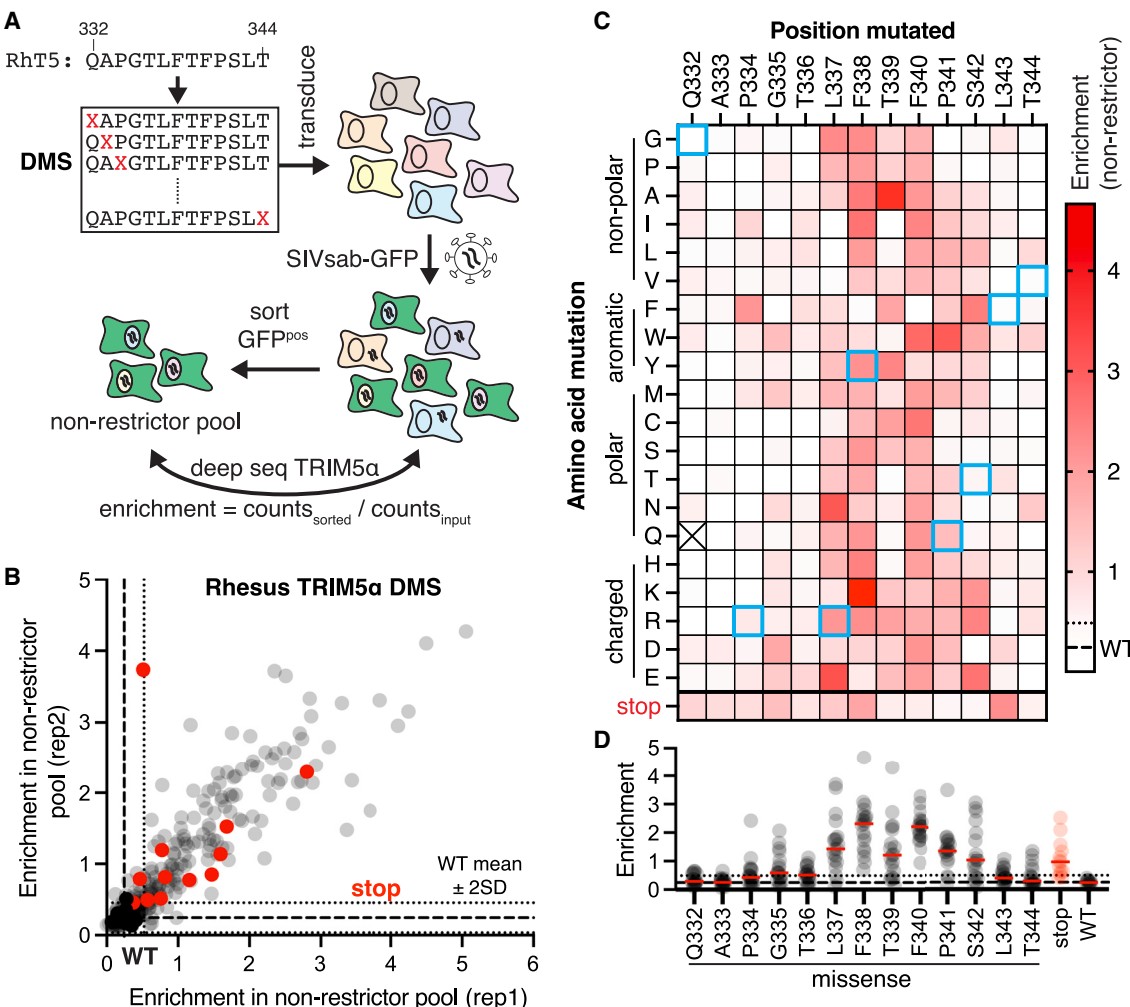

**Figure 3. Most amino acids in central v1 loop positions are incompatible with SIVsab restriction**

(A) CRFK cells expressing a DMS library of rhesus TRIM5α were infected with SIVsab-GFP, and GFP^pos cells were sorted to identify loss-of-function variants ($n = 2$ independent experiments).

(B) Both missense (gray) and truncated variants (red) are enriched in the non-restrictor pool compared to WT in both biological replicates (black; dashed and dotted lines: WT mean ± 2 SD).

(C) Average enrichment in the non-restrictor pool for each missense variant, represented as a heatmap. Rhesus TRIM5α variants with a single human TRIM5α missense mutation are boxed in cyan.

(D) The median enrichment score for missense mutations at each position (red dash) highlights mutational intolerance at positions 337–342.

See also Data S3.

protein-coding indels genome wide are shorter (<4 co-dons).[15,16] Our DIS library for the human TRIM5α v1 loop thus consists of 396 (22 positions × 18 indel) variants.

Using the same workflow as for the DMS and combinatorial libraries, we screened the DIS library for human TRIM5α variants that gained SIVsab restriction. In stark contrast to the insufficiency of missense mutations (Figure 1), we identified 3 duplication variants that were strongly enriched in the SIVsab-restrictor pool (Figure 4B). We confirmed that these human TRIM5α variants gained SIVsab restriction by individually expressing them in cells (Figure 4C). Two of these variants involve overlapping duplications of 4–6 amino acids. However, the most potent of these restrictive alleles only required the duplication of a single amino

acid (a phenylalanine at residue 339). The restriction conferred by this F339dup mutation was equivalent to that of swapping in the entire rhesus TRIM5α v1 loop, representing 9 independent mutations. Thus, in one evolutionary step (duplication of 3 nucleotides), a single indel mutation can confer a potent antiviral function that would otherwise require the simultaneous acquisition of multiple, rare missense mutations and an indel mutation (Figures 1, 2, and 3). Intriguingly, the F339dup mutation not only confers human TRIM5α with SIVsab restriction but also HIV-1 and SIVcpz, although not SIVmac, restriction while leaving N-MLV (N-tropic murine leukemia virus) restriction unimpaired (Figure 4D). Thus, this indel mutation confers broad gains in antiviral function with no apparent evolutionary cost.

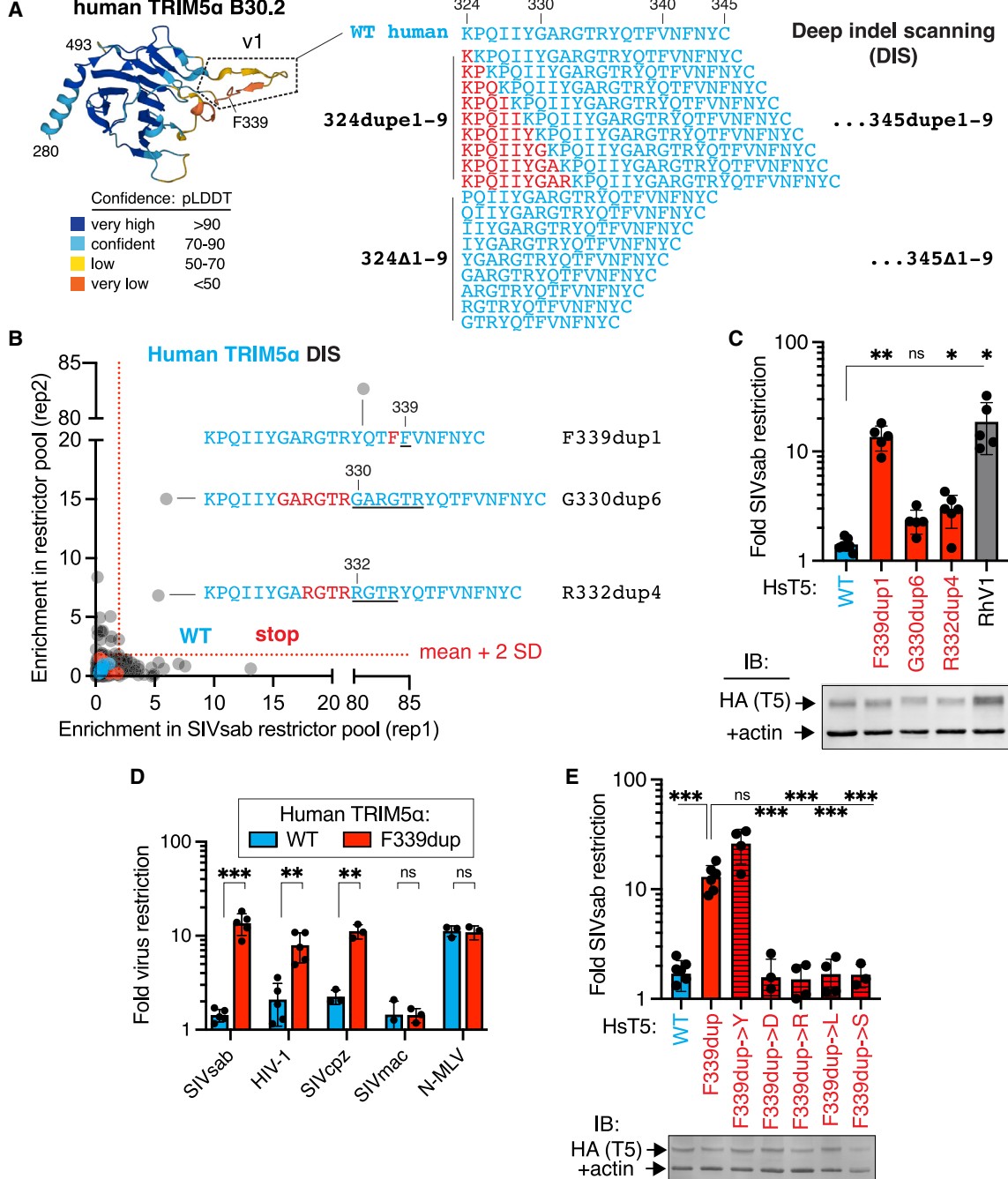

**Figure 4. Human TRIM5α can acquire anti-SIVsab function in a single evolutionary step via indels**

(A) A deep indel scanning (DIS) library of human TRIM5α, comprising in-frame duplications or deletions from 1 to 9 amino acids long at each position in the v1 loop (defined structurally via the AlphaFold[31] TRIM5α model as residues 324–345), was generated and expressed in CRFK cells.

(B) The DIS library was screened for SIVsab restriction (2 biological replicates) as in Figure 1B.

(C) Enriched variants (>[stop mean + 2SD] in both replicates) were individually expressed in CRFK cells and challenged with SIVsab (*n* = 7 [WT], 6 [R332dup4], or 5 [rest] independent experiments).

(D) CRFK cells expressing human TRIM5α with or without the F339 duplication were infected with GFP-marked single-cycle viruses (*n* = 6 [SIVsab], 5 [HIV-1], or 3 [rest] independent experiments).

*(legend continued on next page)*

The F339dup variant is the only single-amino-acid duplication variant we identified as gaining SIVsab restriction (Figure 4B). Moreover, an additional duplication of either of the neighboring residues (T338 or V340) ablated the SIVsab restriction of F339dup (Figure S4B), as did duplicating up to 8 additional residues (Figure 4B). Similarly, missense mutations at 338 or 340 that give rise to 2 tandem phenylalanines do not confer SIVsab restriction (Figure 1). These results suggest that both the length and spacing of this duplication are critical for its functional effect. Moreover, only the insertion of an aromatic residue (F or Y, of 6 residues tested) at this position conferred SIVsab restriction (Figure 4E). Collectively, these results indicate that gain-of-function indel mutations are rare and context dependent but can instantly traverse difficult fitness landscapes for antiviral proteins while providing evolutionary fodder for further refinement (e.g., F339dupY).

### Naturally occurring TRIM5α indels confer new antiviral functions

We detected no obvious hotspots of repetitive sequences that could increase the likelihood of replication slippage in the TRIM5α v1 loop. However, even non-repetitive genomic regions sample indels at ~5% the rate of single-nucleotide polymorphisms,[15] indicating that antiviral proteins like TRIM5α may regularly sample indel variation as a potential source of evolutionary novelty. Indeed, among simian primates, TRIM5α has repeatedly evolved indel mutations, which are strikingly concentrated near sites of repeated missense mutations within the virus-binding B30.2 domain (Figure 5A). Indeed, 3 independent indels have arisen within the ~20-amino-acid specificity-determining v1 loop alone. The first is a 2-residue insertion in the ancestor of macaques, baboons, and mangabeys (*Papionini*); both ancestral and indel TRIM5α variants are encoded by segregating alleles in macaque populations.[32] A more recent 20-codon duplication occurred among African green monkeys, which include sabaeus monkeys (the natural host of SIVsab).[33] Finally, all New World monkeys share a 9-amino-acid deletion,[3] which does not grossly affect their retroviral restriction.[34] This overlapping signature of recurrent indel and missense mutation[33] suggests that indels, like missense mutations, might confer a fitness advantage in rapidly evolving antiviral defense genes.

We directly tested whether these naturally occurring indels confer new antiviral functions by challenging a subset of TRIM5α orthologs with or without the associated indel against a panel of lentiviruses. We found that deleting the 2-amino-acid insertion from rhesus TRIM5α did not impair its ability to restrict HIV-1 but abrogated its SIVsab restriction (Figure 5B). Thus, the naturally occurring, polymorphic 2-residue indel is causally associated with SIVsab restriction in rhesus TRIM5α. Similarly, we found that the 20-amino-acid duplication in sabaeus monkey TRIM5α did not affect its restriction of SIVsab or SIVs circulating in related African green monkeys (grivets

and tantalus monkeys) but was required to restrict SIVmac (Figure 5C), a distinct SIV that evolved in captive macaques.[36] Remarkably, this 20-residue duplication could also confer SIVmac restriction when inserted into the orthologous position of human or rhesus TRIM5α (Figure 5D). Thus, TRIM5α v1 indels that arose during primate evolution are sufficient to confer *de novo* antiviral specificities, demonstrating that indels may have directly contributed to the adaptive evolution of primate TRIM5α in its arms race with viruses. Taken together, our results reveal the remarkable evolutionary potential of indel mutations to confer a restriction factor with new antiviral functions in a single evolutionary leap.

## DISCUSSION

Indels are rare in protein-coding regions compared to the rest of the genome,[15] a strong indication of their deleterious effects on protein function even when they do not result in frameshifts.[14,37] Even when they are retained, indels are overwhelmingly restricted to unstructured regions of proteins (N or C termini, internal disordered loops).[16] This pattern can be explained by the high probability of indels straining or destabilizing structured protein domains.[17,18] However, indels in unstructured loops are frequently tolerated[19–21] and could provide adaptive advantages, enabling proteins to jump to new peaks in fitness landscapes.[38] For example, indels introduced during genetic recombination and later affinity maturation enable B cell receptor genes to encode antibodies with improved antigen affinity.[39] Moreover, in some instances, indels are more likely than missense mutations to improve enzymatic activity[21] or ligand binding affinity,[19] although they are also more likely to compromise these functions.

Here, we add to this emerging paradigm by identifying indels as a critical, often-overlooked source of evolutionary novelty that enables antiviral proteins to overcome viral challenges that cannot be overcome even by multiple missense mutations. We propose that the mechanism by which indels confer gains in antiviral function may be distinct from adaptation through missense mutation. For example, adaptive missense mutations frequently alter charge,[8,28] affecting electrostatic interactions and binding affinity between host and viral proteins. In contrast, our most potent indel variant (F339dup) improves human TRIM5α restriction of both SIVsab and HIV-1 without affecting charge. Moreover, this duplication cannot be phenocopied by missense mutations resulting in 2 tandem phenylalanines, suggesting that its gain of function is not driven by an enthalpically favorable "FF" capsid-binding motif. Instead, we speculate that indels may alter the overall ensemble of conformational states adopted by the largely disordered v1 loop to favor loop poses more compatible with binding the lentiviral capsid.[40] For example, adding a bulky, hydrophobic residue might favor the local packing of

(E) The duplicated residue at position 339 was mutated to biochemically similar (tyrosine, Y) or distinct (aspartate, D; arginine, R; leucine, L; serine, S) residues, expressed in CRFK cells, and challenged with SIVsab (*n* = 3 [R], 4 [D, L, and S], 5 [WT and Y], or 7 [F339dup] independent experiments).

(C–E) Stable expression of all TRIM5α-HA constructs in CRFK was confirmed by immunoblot against HA. Bars, mean; error bars, SD. ns, not significant; *$p < 0.05$, **$p < 0.01$, and ***$p < 0.001$. (C and E) Welch one-way ANOVA with Dunnett's T3 multiple comparisons test vs. (C) WT or (E) F339dupe; (D) Welch's (SIVsab and SIVcpz) or Student's unpaired (all others) two-tailed t test. See also Figure S4 and Data S4.

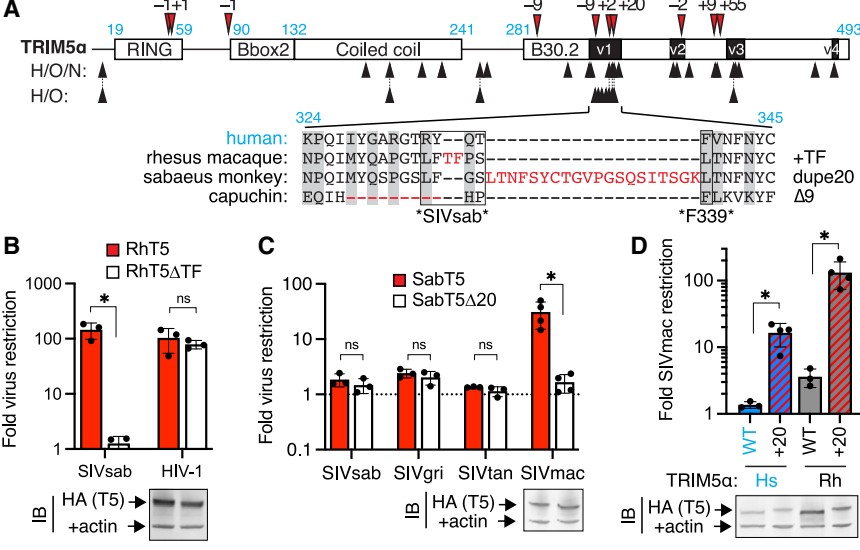

**Figure 5. Naturally occurring primate TRIM5α indels confer new antiviral functions**

(A) TRIM5α sequences from 51 simian primates were aligned, manually curated for in-frame indel mutations (red arrows, amino acid length indicated), and analyzed by PAML[35] (phylogenetic analysis by maximum likelihood) for positions of recurrent missense mutation (black arrows; analyzed for hominoids [H] and Old World monkeys [O] with or without New World monkeys [N] to allow evaluation of the 9 deleted positions in v1). Domain boundaries (human TRIM5α numbering) are indicated in cyan text. Bottom: simian v1 loops have sampled 3 independent indel mutations (red) and 10 positions with recurrent missense mutations (gray boxes). Positions critical for SIVsab restriction by rhesus TRIM5α (Figures 3 and S3) are boxed (*SIVsab*), and the human F339 residue whose duplication confers SIVsab restriction is indicated (*F339*).

(B) Rhesus TRIM5α lacking the TF insertion was stably expressed in CRFK cells and challenged with the indicated viruses (n = 3 independent experiments).

(C) Sabaeus monkey (Sab) TRIM5α, with or without its native 20-amino-acid duplication, was expressed in CRFK cells and challenged with viruses circulating in African green monkeys (SIVsab, sabaeus monkeys; SIVgri, grivets; SIVtan, tantalus monkeys) or SIVmac (n = 4 [SIVmac] or 3 [rest] independent experiments).

(D) Human or rhesus TRIM5α were modified to insert the Sab 20-amino-acid duplication at the orthologous position, expressed in CRFK cells, and challenged with SIVmac (n = 3 [WT] or 4 [+20] independent experiments). Stable expression of all TRIM5α-HA constructs was confirmed by immunoblot against HA.

All bars, mean; error bars, SD. ns, not significant; *p < 0.05; Welch's two-tailed t test. See also Data S5, S6, and S7.

the loop against the B30.2 domain core or promote the formation of a local secondary structure within the loop.[25] Such a mechanism could reduce the entropic penalty of ordering the loop upon binding to the capsid, thereby increasing overall affinity. A reduction in entropy is consistent with our finding that the indel promotes a general gain of function against various lentiviruses.

Many antiviral proteins, including TRIM5α,[41] MxA,[4] and MxB,[42] rely on disordered loops or termini for binding to viral proteins. This property may allow other antiviral proteins to tolerate indel mutations in virus-binding domains, facilitating evolutionary leaps toward novel antiviral specificities. Such evolutionary flexibility contrasts with enzymes and other proteins that preferentially use folded, ordered cores for catalysis or ligand binding,[43,44] which are concomitantly less likely to tolerate either missense or indel mutations.[19] However, enzyme catalysis sometimes involves loops whose evolution modulates activity,[45] and even in the context of folded protein domains, indels are rarely tolerated and can confer adaptive benefits.[19,21] Our work reveals that indels represent a high-risk, high-reward evolutionary strategy of adaptation that may enable antiviral proteins to overcome dire challenges imposed by pathogenic viruses.

### Limitations of the study

Here, we show that indel mutations in TRIM5α's virus-binding loop confer novel recognition of lentiviruses, which we infer is due to increased binding to these viral capsids. However, we were unable to directly measure the affinity of TRIM5α variants for assembled capsids, which, even for potent TRIM5α orthologs, has been reported as very low.[46] Thus, the biophysical ba-

sis by which indels confer gains in function remains an open question. In this study, we limited our analysis to non-frameshifting in-phase indel mutations, which insert or delete amino acid(s) at codon boundaries. However, non-frameshifting indels can also occur after the first or second nucleotide of a codon, thereby potentially causing missense mutations adjacent to the inserted or deleted codon(s). These less conservative, out-of-phase indel mutations might also allow evolutionary leaps toward novel antiviral functions, or they might disrupt any potential gains in function from the indel mutation alone. Future DIS studies should include out-of-phase indels, which, by random chance, should constitute 66% of indel variants, to address their functional effects.

### RESOURCE AVAILABILITY

#### Lead contact
Further information and requests for resources and reagents should be directed to and will be fulfilled by the lead contact, Jeannette L. Tenthorey (jeannette.tenthorey@ucsf.edu).

#### Materials availability
All unique/stable reagents generated in this study are available from the lead contact without restriction.

#### Data and code availability
Illumina sequencing data (adapter trimmed, paired-end-read merged, quality score filtered, and collapsed to number of reads for each unique sequence) and scripts for Illumina data analysis generated in this study are available at Zenodo (https://doi.org/10.5281/zenodo.14898088). All other source data are available in the supplemental information (Data S1, S2, S3, S4, S5, S6, S7, and S8).

## Article

## ACKNOWLEDGMENTS

We thank Janet Young for assistance with scripting for data analysis and all Tenthorey, Emerman, and Malik lab members for constructive feedback. This work was supported by funding from the Howard Hughes Medical Institute, Hanna H. Gray Fellowship GT11096/GT16732 (J.L.T.) and Investigator award (H.S.M.); the G. Harold and Leila Y. Mathers Foundation (H.S.M.); the National Institutes of Health, National Institute of Allergy and Infectious Diseases grant U54 A170792 (M.E. and H.S.M.); and the American Foundation for AIDS Research, Mathilde Krim Fellowship in Biomedical Research #110298-71-RKHF/110537-74-RKHF (J.L.T.).

## AUTHOR CONTRIBUTIONS

Conceptualization, J.L.T. and H.S.M.; methodology, J.L.T.; investigation, S.d.B., I.R., H.K., C.L., and J.L.T.; visualization, J.L.T.; funding acquisition, J.L.T., M.E., and H.S.M.; project administration, J.L.T.; supervision, J.L.T., M.E., and H.S.M.; writing – original draft, J.L.T.; writing – review & editing, M.E. and H.S.M.

## DECLARATION OF INTERESTS

The authors declare no competing interests.

## STAR★METHODS

Detailed methods are provided in the online version of this paper and include the following:

- KEY RESOURCES TABLE
- EXPERIMENTAL MODEL AND SUBJECT DETAILS
  - Cell lines and cell culture
  - Retroviral strains
- METHOD DETAILS
  - Plasmids and cloning
  - TRIM5α variant library construction
  - Virus production and infection
  - Calculation of TRIM5α-mediated restriction
  - Immunoblot
  - Library screens for gain- or loss-of-function TRIM5α variants
- QUANTIFICATION AND STATISTICAL ANALYSIS
  - Illumina data analysis
  - Statistics
  - TRIM5α indel annotation and PAML

## SUPPLEMENTAL INFORMATION

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

🔓 CellPress

**Cell Genomics**
Article

## STAR★METHODS

### KEY RESOURCES TABLE

| REAGENT or RESOURCE | SOURCE | IDENTIFIER |
|---|---|---|
| **Antibodies** | | |
| PE-conjugated anti-HA tag (mouse monoclonal HA.11) | Biolegend | Cat # 901517; RRID:AB_2629622 |
| Anti-HA tag (rabbit monoclonal C29F4) | Cell Signaling Technologies | Cat # 3724; RRID:AB_1549585 |
| Anti-β-actin (rabbit polyclonal) | Abcam | Cat # ab8227; RRID:AB_2305186 |
| IRDye 800CW donkey anti-rabbit | LI-COR | Cat # 926–32213; RRID:AB_621848 |
| **Bacterial and virus strains** | | |
| NEBStable chemically competent cells | NEB | Cat #C3040 |
| Endura electrocompetent cells (for libraries) | Lucigen | Cat # 60242-1-LU |
| **Chemicals, peptides, and recombinant proteins** | | |
| *Trans*-IT 293T transfection reagent | Mirus Bio | Cat # MIR 2700 |
| DEAE-Dextran | Sigma | Cat #D9885 |
| Puromycin dihydrochloride | Fisher | Cat # 50488918 |
| **Critical commercial assays** | | |
| PureYield miniprep kit (transfection-grade DNA) | Promega | Cat # A1223 |
| Nucelobond Xtra midiprep kit (transfection-grade DNA) | Takara | Cat # 740410.50 |
| Cytofix/Cytoperm fixation/permeabilization kit | BD | Cat # 554714 |
| DNeasy Blood and Tissue kit | Qiagen | Cat # 69504 |
| Qubit dsDNA HS assay kit | Invitrogen | Cat #Q32854 |
| **Deposited data** | | |
| Raw Illumina read files and processed data | This manuscript | https://doi.org/10.5281/zenodo.14901870 |
| **Experimental models: Cell lines** | | |
| HEK-293T/17 (human cell line) | ATCC | Cat # CRL-11268; RRID:CVCL_1926 |
| CRFK (cat fibroblast cell line) | ATCC | Cat # CCL-94; RRID:CVCL_2426 |
| **Oligonucleotides** | | |
| Primers used for cloning TRIM5α variants or variant libraries, see Table S1 | This manuscript | Table S1 |
| Primers for Illumina library generation, see Table S2 | This manuscript | Table S2 |
| **Recombinant DNA** | | |
| Plasmid: pMD2.G (VSV-G pseudotyping envelope vector for VLP production) | Gift from Didier Trono (unpublished) | RRID:Addgene_12259 |
| Plasmid: pALPS-eGFP (GFP-encoding HIV-1 transfer vector for VLP production) | McCauley et al.[47] | RRID:Addgene_101323 |
| Plasmid: pHIV-SAB (HIV-1 gagpol expression vector with SIVsab CA for SIVsab VLP production) | Kratovac et al.[29] | N/A |
| Plasmid: pQCXIP-Human TRIM5α-HA (retroviral vector for constitutive expression of human TRIM5α) | Tenthorey et al.[28] | N/A |
| Plasmid: pQCXIP-Rhesus TRIM5α-HA (retroviral vector for constitutive expression of rhesus TRIM5α) | Tenthorey et al.[28] | N/A |
| Plasmid: pQCXIP-Sabaeus TRIM5α-HA (retroviral vector for constitutive expression of sabaeus TRIM5α) | This manuscript | N/A |
| **Software and algorithms** | | |
| Cutadapt | Martin[48] | https://cutadapt.readthedocs.io/en/stable/ |
| NGmerge | Gaspar[49] | https://github.com/jsh58/NGmerge |
| FASTX-toolkit | Hannon[50] | https://github.com/agordon/fastx_toolkit |

*(Continued on next page)*

**Continued**

| REAGENT or RESOURCE | SOURCE | IDENTIFIER |
|---|---|---|
| R | R Core Team[51] | https://www.r-project.org/ |
| Scripts to analyze Illumina libraries | This manuscript | https://doi.org/10.5281/zenodo.14901870 |
| PAML | Yang et al.[35] | http://abacus.gene.ucl.ac.uk/software/paml.html |

## EXPERIMENTAL MODEL AND SUBJECT DETAILS

### Cell lines and cell culture

The following cell lines were purchased from the ATCC: HEK293T/17 (Cat #CRL-11268; RRID:CVCL_1926), a human female fetal cell line that is highly transfectable for high-titer virus production; and CRFK (Cat #CCL-94; RRID:CVCL_2426), a female cat fibroblast cell line used for stable expression of TRIM5α variants followed by retroviral infection to assess their antiviral function. Cell lines were used within 20 passages of receipt from ATCC and not further authenticated after purchase. Cell supernatants were tested for mycoplasma contamination by MycoProbe Kit (R&D Systems) or MycoStrip (InvivoGen). Cells were grown on tissue-culture treated plates in DMEM containing high glucose and L-glutamine (Gibco) and supplemented with 1x penicillin/streptomycin (Gibco) and 10% fetal bovine serum (Gibco). Cells were grown at 37°C in 5% $CO_2$ in humidified incubators. Cells were harvested from plates by digestion with 0.05% trypsin-EDTA (Thermo Fisher) and counted using a TC20 (Biorad) or Vi-Cell BLU (Beckman) automated cell counter.

### Retroviral strains

All retroviral infections were performed using VSV-G-pseudotyped single-cycle virus-like particles (VLPs) that allow only one round of infection (see plasmids and cloning section below for details). All lentiviral infections (HIV or SIV) were made using HIV-1 strain NL4-3 VLPs. As indicated, the capsid sequence was swapped for that of HIV-2 ROD strain, SIVcpz GAB2 strain, SIVmac 239 strain, SIVsab SAB1 strain, SIVgri GRI1 strain, and SIVtan TAN1 strain. Alternately, N-tropic murine leukemia virus (N-MLV) was produced using MLV VLPs.

## METHOD DETAILS

### Plasmids and cloning

Single-cycle viruses were generated from three plasmids: two plasmids for transient expression of the VSV-G pseudotyping envelope protein (pMD2.G, a gift from Didier Trono, Addgene plasmid #12259) and the retroviral gag-pol (see subsequent description), and a virus-matched transfer vector encoding GFP (pALPS-eGFP for HIV/SIV, a gift from Jeremy Luban,[47] Addgene plasmid # 101323; pQCXIP-eGFP[28] for MLV). This three-plasmid design produces virions that integrate and express only GFP in target cells, preventing subsequent rounds of virus production. Viral gag-pol expressing plasmids included HIV-1 NL4-3 strain (p8.9NdSB bGH BlpI BstEII[52]) and vectors that replaced the HIV-1 capsid residues ∼1–200 (Figure S5) with those of: HIV-2 ROD strain (p8.9NdSB bGH BlpI BstEII HIV-2 CA[53]), SIVcpz GAB2 strain (pHIV-Gb2[29]), SIVmac 239 strain (pHIV-MAC with an A77V mutation in the capsid[29]), SIVsab SAB1 strain (pHIV-SAB[29]), SIVgri GRI1 strain, and SIVtan TAN1 strain. The nucleotide sequences for GRI1 (GenBank M29973.1) and TAN1 (GenBank U58991.1) were cloned to replace the HIV-1 CA residues 1–200 within p8.9NdSB bGH BlpI BstEII. The designed sequence between the BlpI and BstEII sites (including cut sites and 4 flanking nucleotides) was synthesized as a gBlock (IDT) and ligated into BlpI/BstEII-digested p8.9NdSB bGH BlpI BstEII. MLV gag-pol expressing plasmids included pCIG3N (used to generate N-tropic MLV-GFP viruses) and JK3 (used to generate TRIM5α transducing viruses). We note that in JK3, gag-pol is driven by the HIV-1 LTR and is therefore Tat-dependent, likely contributing to low (but usable) viral titers in our experiments that did not co-transfect a plasmid for transient expression of Tat.

All TRIM5α constructs were generated with a C-terminal HA epitope tag and cloned into pQCXIP (Takara) to allow CMV-driven TRIM5α expression and selection of transductants via the IRES-puromycin resistance cassette. Human and rhesus TRIM5α, as well as human TRIM5α variants (rhesus v1 chimera; G330E; R332P; R332E; G333D; G333Y; R335E; Q337D; Q337N; V340H) were previously reported.[28] A silent XhoI site 5′ of the v1 loop was engineered into human (codons 320–322) or rhesus TRIM5α (codons 322–324) by Quikchange PCR using primers containing the desired mutations flanked by perfect homology on each side (∼20 nucleotides, ∼52°C melting temperature); Primestar HS polymerase (Takara) was used to minimize errors during full-plasmid amplification (98°C 10''; 24 x [98°C 10″, 62°C, 15″, 72°C, 8.5']; 72°C, 20′), followed by DpnI digestion of unmodified parental DNA. Further targeted TRIM5α mutations were introduced by Quikchange PCR, or chemically synthesized and cloned into the XhoI and EcoRI sites of the appropriate parent (Genscript). To generate a rhesus TRIM5α bearing the human v1 loop, codons 332–344 were replaced with human codons 330–340 by inverse PCR, using outward-facing primers with human TRIM5α overhangs, followed by DpnI digestion and blunt end ligation using KLD enzyme mix (NEB). Sabaeus TRIM5α (XM_008019878.2), with codons 341–360 recoded or deleted and a silent BstB1 site engineered at codons 296–298, was synthesized as a gBlock and ligated between the NotI and EcoRI sites of pQCXIP. Codons 341–360 of sabaeus TRIM5α were inserted into human or rhesus TRIM5α before codon 339 or 343, respectively, by

gBlock synthesis and ligation between the engineered internal XhoI site and flanking EcoRI site of wildtype TRIM5α. All plasmids were cloned into chemically competent NEB-5α or NEBStabl (NEB). Plasmids were purified using PureYield miniprep kits and quantified by NanoDrop A260, and the TRIM5α open reading frame was verified by Sanger sequencing using pQCXIP-F (acaccgggaccgatccag) and internal primers (HsT5-midF: GATCTGGAGCATCGGCTG, RhT5-midF: CTCATCTCAGAACTGGAGCATC; CsabT5-midF: tggagcatcggttgcagg).

### TRIM5α variant library construction

Deep mutational scanning libraries of the v1 loop in human or rhesus TRIM5α were previously reported.[28] All other variant libraries were synthesized as oligo pools (NEB) covering the TRIM5α sequence between the engineered internal XhoI site and the flanking EcoRI site (see Table S1 for ordered sequences).

For the library converting the rapidly evolving v1 loop of human (codons 330–340) and rhesus (codons 332–344) TRIM5α, mixed bases were used to encode both the human and rhesus amino acid where possible, and separate oligos were designed to cover variation that required 2 nucleotide changes or a 6-nucleotide insertion. In total, 8 oligos were designed and pooled for each library, and the v1-flanking sequences of human or rhesus TRIM5α were added to each oligo. A design error in one oligo resulted in a +1 frameshift error for 12.5% of the human TRIM5α/Rhv1 combinatorial library. In addition, a large proportion of variants lacking the TF insertion unexpectedly contained a −1 frameshift error in codon R332/P334 in both human TRIM5α/Rhv1 and rhesus TRIM5α/HsV1 cloned libraries, presumably due to an oligo synthesis error; the correct versions of these variants had low representation in our library and were frequently excluded from our analysis for minimum count thresholds (see Illumina data analysis section below). Therefore, the "-TF" variants are under-represented in each library.

Deep indel scanning oligos were designed for the extended v1 loop (human sites 324–345) to mimic natural primate TRIM5α indels, which include a deletion of 4 residues into this extended loop and a duplication encompassing 13 residues past it. Tiling by codon, each oligo either deleted or duplicated that codon and up to 8 additional downstream codons. Sequence duplications were recoded with as many silent mutations as possible to minimize synthesis errors and recombination. This design strategy is conservative: it avoids the 67% of indels that would cause frameshifts, and it does not allow out-of-codon-phase (but still in-frame) indels that, by virtue of inserting at the second or third nucleotide of a codon, would give rise to adjacent missense mutations.[16] In total, 396 human TRIM5α indel variant oligos were synthesized, with v1-flanking sequences extended to the XhoI and EcoRI sites, as an oligo pool. The pool also included 7 premature stop codon variants and 7 recoded wild-type variants as internal controls. For this library, all designed variants were detected in the final libraries.

Oligo pools were converted to double-stranded DNA by reverse-strand synthesis using Q5 polymerase (NEB) and the reverse primer only (human: gatCAGGATCCAAGCAGTTTTC; rhesus: gatCAggatccAAGCACTTTTC) with four cycles of extension (98°C 5′, 4x [98°C, 15″, 64°C, 30″, 72°C, 30″], 72°C 5′). Reverse strand synthesis was verified by ethidium bromide staining of dsDNA on a 2% agarose gel. Double-stranded libraries were digested with XhoI and EcoRI, gel purified, ligated at 10x molar ratio into human or rhesus TRIM5α cut with XhoI and EcoRI, and transformed into Endura electrocompetent cells (Lucigen) following manufacturer's instructions. Serial dilutions were plated to count the number of unique colonies, and transformations were repeated until at least 100x library coverage was achieved (∼5x10^4 CFU). To assess library quality, 20 random colonies were sequenced from each library. Libraries were harvested by scraping colonies from transformation plates and directly purifying plasmids (with no outgrowth that could introduce amplification bias) using Nucleobond Xtra midiprep kits (Takara).

### Virus production and infection

To generate single-cycle virus, HEK-293T/17 were seeded at 3.5 x 10^5 cells/mL with 2 mL/well in 6-well plates and transfected the following day with 3 plasmids (1 μg of transfer vector, 667 ng of virus-matched gag-pol, and 333 ng of pMD2.G per well) using 3 μL of *Trans*-IT 293T transfection reagent (Mirus Bio) per μg of DNA. HIV and SIV GFP-reporter viruses were produced by co-transfecting pALPS-eGFP, pMD2.G, and the appropriate gag-pol-expressing plasmid. N-MLV GFP-reporter viruses were produced with pQCXIP-eGFP, pCIG3N, and pMD2.G. TRIM5α-transducing virus was produced with the appropriate pQCXIP-TRIM5α construct, JK3, and pMD2.G. After 12–24 h, media was aspirated and replaced with 1 mL of fresh media. Virus-containing supernatants were harvested at 48 h post-transfection, clarified at 500 x g for 5 min to remove cell debris, and snap frozen in liquid nitrogen for storage at −80°C.

To titer viruses or assess TRIM5α-mediated restriction, CRFK cells were seeded at 1 x 10^5 cells/mL with 100 μL/well in 96-well plates the day prior to transduction. Freshly thawed viruses were serially diluted (2- or 3-fold dilution series) in media containing 10 μg/mL polybrene (for MLV infections) or 20 μg/mL DEAE-dextran (for HIV or SIV infections). CRFK media was aspirated and replaced with diluted virus at ½ x volume (i.e., 50 μL/well). Plates were centrifuged at 1100 x g for 30 min at 30°C. The following day, virus was removed and fed fresh media. Infection was monitored by flow cytometry at 72 h post infection to assess GFP (unfixed cells) or TRIM5α-HA expression (using the CytoFix/Cytoperm kit [BD] according to manufacturer's instructions and staining with anti-HA-PE [Biolegend] diluted 1:200. Cells were gated on size (FSC vs. SSC), then single cells (FSC height vs. area), and then GFP+ or HA+ (as compared to uninfected controls; FITC vs. PE).

To stably transduce TRIM5α variants, we chose an MOI of ∼0.25 (25% HA+) to minimize multiple transductions per cell (∼10% of transductants, assuming a Poisson distribution of infection with independent viral particles). CRFK cells were seeded at 1 x 10^5 cells/mL with 1 mL/well in 12-well plates the day prior to transduction; for transduction of libraries, sufficient cells were seeded to generate

at least 100x independent transductions per variant assuming 25% survival. Cells were transduced as in 96-well plates, but with 500 μL of virus per well at the appropriate MOI. At 24 h post-transduction, viral supernatant was replaced with fresh media containing 5 μg/mL puromycin; media was replaced with fresh puromycin-containing media every 2–3 days, and cells were passaged into larger well format as needed, until all untransduced cells had died (5–7 days). Selected cells were then pooled and maintained in 2 μg/mL puromycin (Fisher). All passages maintained at least $5 \times 10^5$ cells (1000x library coverage) to avoid bottlenecking library diversity.

### Calculation of TRIM5α-mediated restriction

Virus inhibition was calculated by comparing infection of CRFK cells expressing empty vector (pQCXIP) or TRIM5α in the same experiment. Cell lines (other than empty vector and the relevant wild-type TRIM5α ortholog) were blinded with a numerical code, which was decoded after data analysis. Serially diluted GFP-marked virus was used to infect cells as above, and % infection was determined by flow cytometry for GFP expression. Wells with <0.5% or >50% GFP+ cells were excluded due to noise or saturation, respectively. A linear regression (against log-transformed data) was then used to calculate the viral dose corresponding to 5% infection ($ID_5$) for each cell line. The $ID_5$ for cells expressing TRIM5α was normalized to empty-vector-expressing cells to yield fold restriction. All fold inhibitions were measured in at least three independent experiments (distinct samples of each TRIM5α-expressing cell line), each of which was performed either in biological singlicate or duplicate. No experimental data (within the linear range) were excluded unless specific experimental errors (e.g., pipette failure) were noted during the experiment.

### Immunoblot

To confirm expression of stably transduced TRIM5α, CRFK cells expressing TRIM5α were harvested by trypsin and counted. A total of $2 \times 10^6$ cells were pelleted, washed in 1mL of PBS, and lysed in 50 μL of lysis buffer (50 mM Tris, pH 8, 150 mM NaCl, 1% Triton X-100, 1x cOmplete EDTA-free protease inhibitor cocktail [Roche]) at $4°C$ for 15 min. Lysates were clarified (21,000 x g, 15 min, $4°C$). Supernatant protein concentration was quantified by Bradford (BioRad) and normalized to load equal protein across all samples (usually ~20 μg per lane). Samples were boiled for 5 min in Laemmli Sample Buffer (Biorad) supplemented with 5% β-mercaptoethanol and loaded onto Mini-PROTEAN TGX stain-free gels (BioRad). Gels were run in Tris/Glycine/SDS buffer (BioRad) for 50 min at 150 V, then transferred semi-dry for 7 min at 1.3 mV using Trans-Blot Turbo 0.2 mm nitrocellulose transfer packs (BioRad). Blots were blocked with Odyssey blocking buffer, then probed with rabbit anti-HA at 1:2000 (RRID:AB_1549585, Cell Signaling Technologies) and rabbit anti-β-actin at 1:5000 (RRID:AB_2305186, Abcam), washed in TBST, and probed with with IRDye 800CW donkey anti-rabbit (RRID:AB_621848, LI-COR) at 1:10,000. All antibodies were diluted in TBST with 5% bovine serum albumin. After washing, blots were imaged at 680 and 800 nm. Bands at appropriate sizes compared to ladder (Precision Plus, Biorad) were cropped and presented in figures; see Data S8 for uncropped images.

### Library screens for gain- or loss-of-function TRIM5α variants

Prior to screening, infection parameters were optimized to ensure minimal sorting of wild-type variants (~1% GFP[neg] human TRIM5α, corresponding to >99% infection of empty-vector expressing cells, for gain-of-restriction screens; ~0.5% GFP[pos] rhesus TRIM5α, corresponding to ~60% infection of empty-vector expressing cells, for loss-of-restriction screens). These conditions resulted in 3–10% of library cells being sorted as GFP[neg] or GFP[pos], respectively. CRFK cells expressing a TRIM5α library were seeded in 12-well plates at $1 \times 10^5$ cells/well the day prior to infection. At least 24 wells ($2.4 \times 10^6$ cells) were seeded to ensure >500x coverage among sorted cells (assuming at least 2 doublings before sorting 4 days after seeding) for each replicate. The following day, cells were infected with SIVsab-GFP (500 μL/well) and spinoculated (1100 x g, 30 min, $30°C$); control infections of empty vector and wild-type TRIM5α-expressing cells were performed in parallel to monitor infection efficiency. The following day, cells were fed fresh media and allowed to grow an additional 2 days before harvesting by trypsinization. Harvested cells were pelleted, resuspended at $5 \times 10^6$/mL, and filtered (0.7 μm) to minimize aggregation. Cells were FACS sorted using stringent gating parameters, set based on uninfected controls, as follows: cell size (FSC vs. SSC), then single cells (FSC height vs. area), then GFP[pos] or GFP[neg] (FITC vs. PE empty channel). At least $5 \times 10^5$ cells (1000x library coverage) were sorted for each replicate. For gain-of-SIVsab restriction screens, sorted GFP[neg] cells were plated, allowed to recover and expand for 3 days, and then re-seeded and infected for a second round of GFP[neg] sorting to enrich true restrictors from cells uninfected by chance. Sorted cells were pelleted and resuspended in PBS, and their genomic DNA was harvested using DNeasy Blood and Tissue kits (Qiagen). Input samples were harvested from infected but unsorted cells for each replicate, and genomic DNA was extracted.

Illumina libraries were constructed from genomic DNA by 2-step PCR amplification using Q5 DNA polymerase (see Table S2 for oligo sequences for PCR-based sequencing library construction). PCR1 amplified the v1 loop of TRIM5α and added adapters, and PCR2 appended a unique 8-base-pair i7 Nextera barcode for each sample as well as P5 and P7 adapters for flow cell binding. For each library, we amplified TRIM5α from 4 samples: 2 input and 2 sorted replicates. Each sample contained 300 ng of gDNA in each of 5 PCR tubes to avoid PCR jackpotting, thus sampling a total of 1.5 μg of DNA (~500x library coverage, assuming 6.6 pg DNA/cell and a single TRIM5α integration/cell). Products from PCR1 (98°C, 2'; 18 x [98°C, 15"; 70°C, 30"; 72°C, 15"]; 72°C, 30") were digested with ExoI (15' at 37°C) to remove primers, pooled from pentuplicate tubes, purified by Qiaquick purification kit, and split between 4 PCR tubes for PCR2 (98°C, 2'; 15 x [98°C, 15"; 67°C, 30"; 72°C, 15"]; 72°C, 30").

Barcoded PCR products were pooled from quadruplicate tubes and purified by double-sided size selection using Ampure beads (Beckman Coulter). Large DNA was first removed by incubation with 0.8x bead volume and magnetization of beads. The supernatant

was then incubated with 1.4x bead volume to bind the ∼300-base-pair PCR product. Beads were washed with 80% ethanol, and DNA was eluted in water. PCR products were quantified by Qubit DNA HS kit, pooled at equimolar ratios, and size-verified by Tapestation. Pooled samples were Illumina sequenced with paired-end 150-base-pair reads on either NovaSeq 6000 or HiSeq instruments (Novogene) using standard Nextera sequencing primers. Read depth always exceeded 1 million counts per sample (>1000x read coverage of library size).

## QUANTIFICATION AND STATISTICAL ANALYSIS

### Illumina data analysis

Illumina reads were trimmed to remove TRIM5α-homology primer sequence (and, by extension, terminal adapters) from both 5′ and 3′ ends using CutAdapt.[48] Paired-end reads were merged to generate a consensus sequence using NGmerge[49] in dovetail mode to remove 3′ overhangs. Using the FASTX-toolkit,[50] low-quality consensus reads were filtered for a minimum score of Q25 at each nucleotide, and remaining reads were collapsed to unique sequences and counts per sequence.

For DMS libraries, the resulting fasta files were read into R[51] and merged based on sequence into a single table containing the number of read counts for each unique sequence across all conditions (4 per library: 2 input and 2 sorted conditions). Sequences were translated and identified for mutated codon(s) relative to wild-type TRIM5α. Sequences were filtered for expected length (33 nucleotides for human TRIM5α, 39 nucleotides for rhesus TRIM5α), ≤1 mutated codon, and encoding a C or G in the codon wobble position (libraries were created by NNS codons). A pseudocount of 1 read was added to each sequence in each sample to avoid dividing by (or returning) 0. Read counts were then normalized to the total number of reads for each sample. Sequences with <10 counts per million in either input library were removed for having too much noise in subsequent analysis. Enrichment was calculated for each sequence by dividing sorted counts per million (cpm) by input cpm for each replicate. All synonymously-coding variants were averaged to yield an enrichment score for each protein sequence for each replicate, and the standard deviation of enrichment was calculated across synonymous variants and replicates, with one exception: wild-type synonymous variants were plotted separately to give a visual representation of the standard deviation in each replicate. Fully wild-type nucleotide sequence was removed from further analysis because (1) it was a strong outlier relative to all other synonymous variants, and (2) this effect could have been due to plasmid contamination of samples with wild-type TRIM5α.

All other libraries were processed in the same manner, except that instead of filtering as above, fasta files were mapped to a table of expected sequences based on library construction, and only sequences matching these exact nucleotide sequences were retained. Typically, >90% of reads matched these expected sequences. However, for the combinatorial library converting human into rhesus TRIM5α, we noticed a large fraction (∼25%) of reads being lost at this step. Subsequent analysis identified two frameshift mutations (see variant library construction section above), which were then included (as combinatorial variants) in the expected sequence list. After this correction, >90% of reads matched expected sequences. Fully wild-type sequence was retained for this library, as we failed to include wild-type sequences in our library design; however, it was removed from analysis of the indel library, which was designed with 7 wild-type-synonymous variants.

Gain-of-function cut-offs were set based on either premature stop codons (for DMS or indel scanning libraries; > [mean +2 SD] in each replicate) or frameshift variants (for combinatorial human-to-rhesus library; > max score in each replicate). This choice was both more conservative for detecting any antiviral function, and it also frequently included more controls to give a better estimate of assay noise. Loss-of-function cut-offs were based on wild-type rhesus TRIM5α variants (>[mean +2 SD] in each replicate).

### Statistics

Statistics were performed with GraphPad Prism. Prior to statistical analysis, data were tested for normal (Gaussian) distribution by the Shapiro-Wilk test and equal variance (standard deviation). Where all groups to be compared passed the normality test ($\alpha = 0.05$) and had similar variance, mean values were compared by student's unpaired 2-tailed t tests (for two groups) or one-way ANOVA with Dunnett's T3 multiple comparisons test (for multiple groups). For normally distributed data with unequal variance across groups, mean values were compared by Welch's 2-tailed t test (for two groups; unequal variance confirmed by F test) or Welch one-way ANOVA with Dunnett's T3 multiple comparisons correction (for multiple groups). For comparisons where at least one group did not pass normality, mean values were compared by Kruskal-Wallis test with Dunn's multiple comparisons correction (for multiple groups). Statistical details (number of replicates, test used, definition of center, and measures of dispersion) can be found in the relevant figure legends.

To compare positional sequence preference for combinatorial libraries, we generated a 2x2 matrix for the total number of variants (i) with either human or rhesus sequence at that position, (ii) in either the sorted or input pools. Matrices were subjected to a Chi-square test in R.

### TRIM5α indel annotation and PAML

A BLASTn search using human TRIM5α (NM_033034.3) as a query against the nt database (restricted to simian primates and excluding human sequences) returned sequences from 54 species. The reference sequence (or where no reference was available, either the longest isoform or highest-scoring hit) from each species was selected for further analysis, and TRIMCyp genes (from *Aotus* and *Macaca* genera) were excluded. Two sequences were further excluded as low quality (*Rhinopithecus bieti*, XM_017887309.1) or

too highly diverged from other close relatives for high confidence (*Alouatta sara*, AY843511.1). Open reading frames were extracted from the 51 remaining sequences (see Data S6 for accessions) and translation aligned using MUSCLE, with manual adjustments as needed (see Data S5 for final alignment). Indel positions present in any lineage were called manually for position and amino acid length. PAML[35] was used to detect sites of recurrent missense mutation, as previously described.[3] In brief, a PHYML tree was built for alignments with or without New world monkeys (to allow for PAML analysis of the deleted residues from this clade) using the HKY85 substitution model, and the unrooted trees were used as PAML input with their respective alignments. Analyses were run using both F3x4 and F61 codon models to ensure robust results. Residues were defined as rapidly evolving if, under model 8 (beta distribution with positive selection allowed), their Bayes Empirical Bayes values exceeded 0.95 using both codon models.

