## [Document S2. Transparent peer review records for Tenthorey et al · Cell Genomics]

Summary

Initial submission: Received : 9/26/2024

Scientific editor: Laura Zahn

First round of review: Number of reviewers: 4
Revision invited : 10/28/2024
Revision received : 12/21/2024

Second round of review: Number of reviewers: 3
Accepted :2/25/2025

Data freely available: Yes

Code freely available: Yes

This transparent peer review record is not systematically proofread, type-set, or edited. Special characters, formatting, and equations may fail to render properly. Standard procedural text within the editor's letters has been deleted for the sake of brevity, but all official correspondence specific to the manuscript has been preserved.

Referees' reports, first round of review

Reviewer #1: The authors perform saturation substitution, indel and combinatorial mutagenesis of a loop of the human anti-retroviral protein TRIM5alpha and select for its ability to restrict a divergent simian immunodeficiency virus. The data are very clear: more than 5 missense variants observed in evolution are required to confer resistance but a single 1 AA duplication is also sufficient. Overall I think this is a very clear and thought-provoking study that provides a beautiful example of how insertions can drive protein innovation. The data are strong, support the conclusions and are very clearly presented. I strongly recommend acceptance without delay with only very minor modifications.

Minor comment

In figure 1C there is a lack of controls in the substitution DMS scan for the experiment technically working. How do the authors know the selection actually worked when there is no dynamic range in the effects of the missense variants? If a positive control was included in the scan it should be included in the figure.

Reviewer #2: Comments enter in this field will be shared with the auTenthorey et al. conducted a large-scale mutation screening to assess the viral specificity of the human TRIM5-alpha protein, focusing on identifying mutations that could confer resistance to SIVsab. The authors first demonstrated that the v1 loop is crucial for determining target specificity, and they confirmed that a substitution in this region enabled resistance to HIV, a finding previously established in earlier research. They then extended their analysis to SIVsab and found that no single amino acid mutation alone could restrict the virus. However, they identified six mutations, including one indel, in the human TRIM5alpha protein that successfully restricted SIVsab infection, albeit with no intermediate genotypes observed. Remarkably, the study showed that even a single amino acid duplication was sufficient to grant TRIM5alpha the ability to restrict SIVsab, suggesting that the phenotypic potential of indel mutations may be much greater than that of substitutions.

Major comments,

Understanding how novel functions evolve in proteins is a central question in evolutionary biology. By conducting deep mutation screening of TRIM5alpha, an important viral restriction factor that has been extensively studied, the authors demonstrated that new functions can arise more easily through indels than through substitutions. The results shown by the authors are fascinating. The manuscript is well-written, and the experiments were carefully executed. The results strongly support the authors' conclusions. I have one major and a few minor comments.

Regarding the title, while the authors aim to generalize their findings, the mechanisms of protein-protein interactions in viral restriction via loop structures (such as TRIM5alpha) might differ from evolutionary paths in other proteins (e.g., enzymes gaining new substrate specificity). The authors should elaborate more in the Introduction and Discussion, explaining the mechanisms of TRIM5alpha's viral restriction function and discussing whether similar principles could apply to other non-viral protein-protein interactions. This would prevent readers from drawing misleading conclusions.

Minor comments,

1. Introduction: the term "protein-coding indels" cannot distinguish whether the indel causes frameshift or not. I guess the authors here meant the indels at amino-acid level.
 2. Describe the Linnean name of sabaeus monkey.
 3. In Fig.1A, statistical significance should be shown in the figure or main text.
 4. I'm not sure where it would be most appropriate, but the position of F339 could be highlighted in one of the figures (perhaps Fig. 4 or 5?).
- thor; your identity will remain anonymous.

Reviewer #3: This is a great piece of work that demonstrated that indels (in particular insertions in this case) can provide innovative phenotype of protein in natural setting, especially human

evolution. While many mutational studies have been conducted in many protein systems, indels have been overlooked forms of mutations in our community. It is probably only because of convenience in terms of bioinformatics as well as experimental procedures, but not due to less significance of indels in evolution. Indels can be considered to be highlight deleterious but they can also provide much drastic changes in protein structure and function. The authors elegantly demonstrated this by studying sequence changes in v1 loop of human TRIM5a, a restriction factor that inhibits retroviral infection. They performed a series of mutational study, especially comprehensive high-throughput deep mutational scanning to test the effect of mutations on v1 loop to the sensitivity to SIVsab. The discovery in this manuscript is intriguing; it requires much substitutions to alter the affinity against SIVsab but a single insertion can completely abolish the affinity.

The paper is well written in a very concise manner (which I really like). Experiments are well designed and executed (a series of mutational analyses) which collectively provided a convincing and cohesive story. Figures are well-made, interactive and easy to understand. The manuscript provides an very interesting picture that indels can provide an prompt innovation while substitutions cannot reach quickly. I believe that the manuscript is well deserved in publishing high-profile journal such as Cell Genomics. I have only few suggestions to improve the manuscript.

1. While I like the writing style of this paper, which is very concise, I would appreciate a bit more to description of the background of TRIM5a. For example, what is the protein size, what are the typical length of v1Loop, what is the sequence identity of TRIM5a among primates (mammals) for the entire protein? What are the diversity of v1Loop (among primates and mammals)? Some information comes gradually throughout the manuscript, but it is useful to have those upfront in Introduction or the beginning of Results.

2. In Page 5 (as in PDF pages, there is no page number in the manuscript). They say tested 2^9 combinatorial variants, but characterized only 349 variants (out of 512 potential variants). It is fine to be that way, but it should be described more clearly in the manuscript.

3. The discussion can be a bit more expanded. I would be interested in hearing what are the molecular mechanisms for this single insertion can do but multiple substitutions cannot. The authors described in very general manner, but do they have any insights (or guess) in this particular system? I think it might be worth to extend implications to general protein evolution.

Reviewer #4: Antiviral proteins can evolve to recognize viral proteins or macromolecular shapes in the context of constant virus evolution or spillover events through two mechanisms. First, they may adapt to a new virus through missense mutations, and this has been characterized for several examples, including TRIM5. A second, less characterized mechanism, is changing their recognition interface through indels. Tenthorey et al use TRIM5 as a specific example to show how indels modulate TRIM5 antiviral activity. This is an interesting paper that address a key evolutionary point. The authors have made a convincing case that indels modulate the activity of TRIM5 for restricting specific lentiviruses. They have shown that a single insertion (F339dup) increases human TRIM5 restriction of SIVsab and no individual missense mutation in the v1 loop can do so.

Major comments:

1. While the experiments show that indels modulate TRIM5 antiviral activity against specific retroviruses, there are no binding assays or structural understanding of how these changes affect the ability of TRIM5 to interact with retroviral capsids. Can the interaction between TRIM5 variants and lentiviral capsids be tested to correlate restriction with binding for the most relevant indels or could this be modelled using existing structures? The physical interaction between TRIM5 and viral capsids should be discussed in more detail in this manuscript.

2. A limitation of this paper is that most of the data focuses on changes in the v1 loop in TRIM5, while Figure 2A shows that this is only one determinant for differences between human and rhesus macaque TRIM5 antiviral activity against SIVsab. Based on this figure, the v1 loop appears to be necessary but not sufficient for full antiviral activity. The authors should map the other determinants in rhesus macaque TRIM5. How would these other determinants affect TRIM5

antiviral activity?**Minor comments:**

1. For figure panels that compare fold restriction between different retroviruses (Figures 1A, 4D, 5B-C), the data used to calculate the fold restriction (infectivity for empty vector and the TRIM5 vector) should be shown in the supplementary data to show how the different viruses infect the target cells. The different viruses may have substantially different infection rates for the cells, which is masked when only the fold-restriction is shown.

2. Why does SIVsab pose "a more formidable evolutionary challenge than other lentiviruses" for human TRIM5? This is an interesting question and there does not appear to be a potential answer in the manuscript. Can the authors briefly discuss some possibilities?

Authors' response to the first round of review

Reviewer #1:

The authors perform saturation substitution, indel and combinatorial mutagenesis of a loop of the human anti-retroviral protein TRIM5alpha and select for its ability to restrict a divergent simian immunodeficiency virus. The data are very clear: more than 5 missense variants observed in evolution are required to confer resistance but a single 1 AA duplication is also sufficient. Overall I think this is a very clear and thought-provoking study that provides a beautiful example of how insertions can drive protein innovation. The data are strong, support the conclusions and are very clearly presented. I strongly recommend acceptance without delay with only very minor modifications.

We thank the reviewer for their enthusiasm about our manuscript. We have addressed the suggested minor modifications below.

Minor comment

In figure 1C there is a lack of controls in the substitution DMS scan for the experiment technically working. How do the authors know the selection actually worked when there is no dynamic range in the effects of the missense variants? If a positive control was included in the scan it should be included in the figure.

The reviewer makes an excellent point about the need for controls in screens. We included a variety of negative controls (WT synonymous variants and premature stop codon variants) but not a positive control (such as rhesus TRIM5) in the library for Figure 1C. Nevertheless, we are confident that our assay described in Figure 1C has sufficient dynamic range to uncover the effects of missense variants (if there is an effect) for the following reasons:

- 1. Our DMS of rhesus TRIM5 uncovered many single missense variants that partially or completely lost restriction of SIVsab (Figure 3), highlighting our ability to detect the effects of missense mutations on SIVsab restriction.
- 2. We successfully identified both strong and modest gain-of-function variants against SIVsab from our combinatorial or indel variant libraries of human TRIM5 (Figures 2 and 4), indicating that our assay does select for a gain of anti-SIVsab function with a wide dynamic range.
- 3. In prior work, we successfully identified single missense variants of human TRIM5 that gained restriction of HIV-1, indicating that the assay also works for identifying single missense variants that improve antiviral functions (Tenthorey et al. 2020).

Reviewer #2:

Tenthorey et al. conducted a large-scale mutation screening to assess the viral specificity of the human TRIM5-alpha protein, focusing on identifying mutations that could confer resistance to SIVsab. The authors first demonstrated that the v1 loop is crucial for determining target

specificity, and they confirmed that a substitution in this region enabled resistance to HIV, a finding previously established in earlier research. They then extended their analysis to SIVsab and found that no single amino acid mutation alone could restrict the virus. However, they identified six mutations, including one indel, in the human TRIM5alpha protein that successfully restricted SIVsab infection, albeit with no intermediate genotypes observed. Remarkably, the Response to Reviewers

study showed that even a single amino acid duplication was sufficient to grant TRIM5alpha the ability to restrict SIVsab, suggesting that the phenotypic potential of indel mutations may be much greater than that of substitutions.

Understanding how novel functions evolve in proteins is a central question in evolutionary biology. By conducting deep mutation screening of TRIM5alpha, an important viral restriction factor that has been extensively studied, the authors demonstrated that new functions can arise more easily through indels than through substitutions. The results shown by the authors are fascinating. The manuscript is well-written, and the experiments were carefully executed. The results strongly support the authors' conclusions. I have one major and a few minor comments. We are delighted to hear the reviewer's positive comments. To improve our manuscript, we address the reviewer's comments below.

Major comments:

Regarding the title, while the authors aim to generalize their findings, the mechanisms of protein-protein interactions in viral restriction via loop structures (such as TRIM5alpha) might differ from evolutionary paths in other proteins (e.g., enzymes gaining new substrate specificity). The authors should elaborate more in the Introduction and Discussion, explaining the mechanisms of TRIM5alpha's viral restriction function and discussing whether similar principles could apply to other non-viral protein-protein interactions. This would prevent readers from drawing misleading conclusions.

We agree with this comment. To provide additional context for the generality of our findings, we have added the following text:

☐ “TRIM5 \$\alpha\$ uses its disordered v1 loop to bind the capsid core of incoming lentiviruses, HIV-related retroviruses circulating in hominoid and Old world monkey species” (introduction, page 2, lines 29-30).

☐ “[TRIM5's] evolutionary flexibility contrasts with enzymes and other proteins that preferentially use folded, ordered cores for catalysis or ligand binding, which are concomitantly less likely to tolerate either missense or indel mutations. However, enzyme catalysis sometimes involves loops whose evolution modulates activity, and even in the context of folded protein domains, indels are rarely tolerated and can confer adaptive benefits.” (discussion, page 7, lines 23-27).

Minor comments:

1. Introduction: the term "protein-coding indels" cannot distinguish whether the indel causes frameshift or not. I guess the authors here meant the indels at amino-acid level.

In our revised manuscript, we have clarified our wording to more precisely indicate that selection acts “to remove both frame-shifting and in-frame (multiples of 3 nucleotides) indels from proteincoding sequences. Frameshift indels dramatically alter downstream protein sequence and are likely to be instantaneously deleterious, but even in-frame indels often introduce register shifts or bulges in secondary structural elements that destabilize protein folds” on page 2, lines 17-21.

2. Describe the Linnean name of sabaeus monkey.

This has been added to the introductory text on page 2, line 41.

3. In Fig.1A, statistical significance should be shown in the figure or main text.

We have added appropriate statistical tests to Fig. 1A.

4. I'm not sure where it would be most appropriate, but the position of F339 could be highlighted in one of the figures (perhaps Fig. 4 or 5?).

We have boxed the position of F339 in the alignment of TRIM5 sequences in Fig. 5A. In addition, we have added position numbers to the gain-of-function duplication variants in Fig. 4B, and we indicated the structural position of F339 in the AlphaFold model of TRIM5 in Fig. 4A.

Reviewer #3:

This is a great piece of work that demonstrated that indels (in particular insertions in this case) can provide innovative phenotype of protein in natural setting, especially human evolution. While many mutational studies have been conducted in many protein systems, indels have been overlooked foams of mutations in our community. It is probably only because of convenience in terms of bioinformatics as well as experimental procedures, but not due to less significance of indels in evolution. Indels can be considered to be highlight deleterious but they can also provide much drastic changes in protein structure and function. The authors elegantly demonstrated this by studying sequence changes in v1 loop of human TRIM5a, a restriction factor that inhibits retroviral infection. They performed a series of mutational study, especially comprehensive high-throughput deep mutational scanning to tes the effect of mutations on v1 loop to the sensitivity to SIVsab. The discovery in this manuscript is intriguing; it requires much substitutions to alter the affinity against SIVsab but a single insertion can completely abolish the affinity.

The paper is well written in a very concise manner (which I really like). Experiments are well designed and executed (a series of mutational analyses) which collectively provided a convincing and cohesive story. Figures are well-made, interactive and easy to understand. The manuscript provides an very interesting picture that indels can provide an prompt innovation while substitutions cannot reach quickly. I believe that the manuscript is well deserved in publishing high-profile journal such as Cell Genomics. I have only few suggestions to improve the manuscript.

We are grateful for the laudatory comments of the reviewer. We have taken the reviewer's suggestions for improving our manuscript, as detailed below.

1. While I like the writing style of this paper, which is very concise, I would appreciate a bit more to description of the background of TRIM5a. For example, what is the protein size, what are the typical length of v1Loop, what is the sequence identity of TRIM5a among primates (mammals) fo the entire protein? What are the diversity of v1Loop (among primates and mammals)? Some information comes gradually throughout the manuscript, but it is useful to have those upfront in Introduction or the beginning of Results.

To provide the reader with more context, we have added the following text to the introduction: "The v1 loop is a small segment (typically 22 amino acids) of the virus-binding B30.2 domain of TRIM5, which also contains self-oligomerization domains (coiled-coil, B-box) and a ubiquitin ligase (RING) domain within the ~500 amino acid protein. The name v1 (for "variable region 1") reflects its dramatic divergence across primate TRIM5 proteins, both in sequence (0-95% identity among primate pairs, compared to >70% identity across the entire protein) and length (13-42 amino acids) (see Data S5). (page 2, lines 31-37).

In addition, we have added residue numbers for domain boundaries in Figure 5A.

2. In Page 5 (as in PDF pages, there is no page number in the manuscript). They say tested 2^9 combinatorial variants, but characterized only 349 variants (out of 512 potential variants). It is fine to be that way, but it should be described more clearly in the manuscript.

We apologize for the lack of page numbers and have now added them for easier reference.

In our methods section (TRIM5 variant library construction sub-section, page 28), we describe oligonucleotide synthesis errors that resulted in missing or poorly represented sequences, which

were excluded from our analysis. We have now added a line to our results section after indicating that we analyzed 349 of 512 possible variants, referring readers to the methods section for details on why other variants were excluded (“see STAR Methods: TRIM5 variant library construction for a DNA synthesis error resulting in excluded variants”, page 4, lines 7-8).

3. The discussion can be a bit more expanded. I would be interested in hearing what are the molecular mechanisms for this single insertion can do but multiple substitutions cannot. The authors described in very general manner, but do they have any insights (or guess) in this particular system? I think it might be worth to extend implications to general protein evolution. We have expanded our discussion to speculate further about the molecular mechanism by which the indel variant confers a gain of function, as follows: “Moreover, this duplication cannot be phenocopied by missense mutations resulting in 2 tandem phenylalanines, suggesting that its gain of function is not driven by an enthalpically favorable “FF” capsid-binding motif. Instead, we speculate that indels may alter the overall ensemble of conformational states adopted by the largely disordered v1 loop, to favor loop poses more compatible with binding the SIVsab lentiviral capsid. For example, adding a bulky, hydrophobic residue might favor the local packing of the loop against the B30.2 domain core or promote the formation of local secondary structure within the loop. Such a mechanism could reduce the entropic penalty of ordering the loop upon binding to the capsid, thereby increasing overall affinity. A reduction in entropy is consistent with our finding that the indel promotes a general gain of function against various lentiviruses.” (page 7, lines 10-19)

Reviewer #4:

Antiviral proteins can evolve to recognize viral proteins or macromolecular shapes in the context of constant virus evolution or spillover events through two mechanisms. First, they may adapt to a new virus through missense mutations, and this has been characterized for several examples, including TRIM5. A second, less characterized mechanism, is changing their recognition interface through indels. Tenthorey et al use TRIM5 as a specific example to show how indels modulate TRIM5 antiviral activity. This is an interesting paper that address a key evolutionary point. The authors have made a convincing case that indels modulate the activity of TRIM5 for restricting specific lentiviruses. They have shown that a single insertion (F339dup) increases human TRIM5 restriction of SIVsab and no individual missense mutation in the v1 loop can do so.

We are glad the reviewer finds our manuscript a convincing case for indels modulating TRIM5 activity, which we believe is the most exciting finding in our manuscript.

Major comments:

1. While the experiments show that indels modulate TRIM5 antiviral activity against specific retroviruses, there are no binding assays or structural understanding of how these changes affect the ability of TRIM5 to interact with retroviral capsids. Can the interaction between TRIM5 variants and lentiviral capsids be tested to correlate restriction with binding for the most relevant indels or could this be modelled using existing structures? The physical interaction between TRIM5 and viral capsids should be discussed in more detail in this manuscript.

Although we agree with this comment, unfortunately, a biochemical assay to determine the avidity of TRIM5 variants for lentiviral capsids has eluded the field for decades. Previous attempts have either failed or have not found the predicted correlation between apparent capsid binding (by co-pelleting) and antiviral function. For example, according to the co-pelleting assay, wild-type human TRIM5 appears to bind HIV-1 better than rhesus TRIM5 despite its lack of antiviral activity (Selyutina et al. 2020, PMID: 32187548, see Figure 1). However, multiple groups have structurally implicated the v1 loop in the interface with the lentiviral capsid. Therefore, we fully expect v1 loop changes to alter antiviral function by modulating avidity for

the capsid, even though this has not been formally shown. We have added the following text to the manuscript to clarify these points:

☐ “TRIM5 uses its disordered v1 loop to bind the capsid core of incoming lentiviruses... although the molecular details of this interface remain unclear²⁴⁻²⁷” (introduction, page 2, lines 29-31).

Per the reviewer’s suggestion, we also attempted to use AlphaFold3 to model the interaction between lentiviral capsids and TRIM5 variants. Unfortunately, AlphaFold largely returned nonsensical results (e.g., the TRIM5 B30.2 domain binding on the interior face of the capsid, which is not sterically accessible during infection), reflecting the difficulty of modeling the interaction between TRIM5 and the oligomeric capsid.

2. A limitation of this paper is that most of the data focuses on changes in the v1 loop in TRIM5, while Figure 2A shows that this is only one determinant for differences between human and rhesus macaque TRIM5 antiviral activity against SIVsab. Based on this figure, the v1 loop appears to be necessary but not sufficient for full antiviral activity. The authors should map the other determinants in rhesus macaque TRIM5. How would these other determinants affect TRIM5 antiviral activity?

We focus our manuscript on the v1 loop of TRIM5 for several reasons. It is an evolutionary hotspot of TRIM5 adaptive mutation (Figure 5A), and it has been implicated by multiple groups as being involved in both binding lentiviral capsids and determining lentiviral specificity. Indeed, the data we present here are consistent with the v1 loop being the dominant driver of SIVsab recognition. For example, single missense mutations to a human-mimicking residue within the v1 loop of rhesus TRIM5 completely ablate SIVsab restriction (Figures 3, S3D), as does swapping the entire rhesus TRIM5 v1 loop for that of human TRIM5 (Figure 2A). Thus, we are confident that the v1 loop is the primary determinant of SIVsab recognition for rhesus TRIM5. Nevertheless, the reviewer correctly points out that human TRIM5 does not recover full rhesuslike activity when conferred with the rhesus v1 loop (Figure 2A). These data suggest that the human TRIM5 backbone is not fully competent for antiviral function. Indeed, this result is consistent with previous reports that an HsT5-RhV1 chimera is only moderately functional against HIV-1 (Sawyer et al. 2005, PMID: 15689398, see Figure 1D). It is not yet clear whether this effect is mediated by a single region of large effect (like the v1 loop), which we could identify with our screens, or many regions of small effect. Because our manuscript is focused on the evolution of novel antiviral specificity, rather than on human TRIM5 per se, we believe that mapping these determinant(s) of human TRIM5 incompatibility are beyond the scope of this paper.

To clarify these points for readers, we have added the following text to the manuscript: “The partial functionality of [the human TRIM5 – rhesus v1] chimera is consistent with its modest activity (relative to rhesus TRIM5 α) against HIV-1” (page 3, lines 39-41).

Minor comments:

1. For figure panels that compare fold restriction between different retroviruses (Figures 1A, 4D, 5B-C), the data used to calculate the fold restriction (infectivity for empty vector and the TRIM5 vector) should be shown in the supplementary data to show how the different viruses infect the target cells. The different viruses may have substantially different infection rates for the cells, which is masked when only the fold-restriction is shown.

We agree with the reviewer. We have included the source data in our re-submission via Mendeley (<https://data.mendeley.com/preview/5ks4p56z33?a=e7b82ab3-9eea-47ae-8e54-8bd703af5b03>). In general, these source data show that all viruses are ~equally infectious, except HIV2 (~10-fold lower infectivity than HIV1).

2. Why does SIVsab pose "a more formidable evolutionary challenge than other lentiviruses" for

human TRIM5? This is an interesting question and there does not appear to be a potential answer in the manuscript. Can the authors briefly discuss some possibilities?

The reviewer poses a fantastic question. Unfortunately, we do not have a good understanding of why SIVsab differs from HIV-1, HIV-2, SIVcpz and SIVmac in terms of its evolutionary challenge. We tried the following to attempt to shed light on this question:

1. We used AlphaFold to model the structure of each of these capsids and found them all to be highly similar to HIV-1 (RMSD ≤ 0.85 angstroms), including SIVsab (RMSD = 0.63 angstroms).

2. We did not see any obvious sequence features that distinguish SIVsab from the other lentiviruses we tested that can be more “easily” adapted against (see Figure S5).

3. The CypA-binding loop has been repeatedly implicated in TRIM5-mediated recognition.

To test whether this loop confers the difference in “evolutionary challenge”, we made chimeras in which we swapped the CypA loop between HIV-1 and SIVsab.

Unfortunately, these chimeras yielded non-infectious virus, preventing us from testing this hypothesis.

Ultimately, we agree that this is a very interesting question. Answering it will require us to devise and implement deep mutational scans of different lentiviral capsids. However, we believe it is beyond the scope of the current manuscript, in which we focus on the molecular basis of TRIM5 adaptation.

Referees' reports, second round of review

Reviewer #1: The authors have addressed our concerns - many thanks!

Reviewer #2: The authors have appropriately addressed my comments, and the manuscript now appears suitable for publication.

Reviewer #4: Tenthorey et al have revised the manuscript by providing additional text that address my major comments and have provided the source data on Mendeley. This is an excellent manuscript that provides new insight into how antiviral proteins evolve.

Authors' response to the second round of review